# Mushrooms as a Resource for Mibyou-Care Functional Food; The Role of Basidiomycetes-X (Shirayukidake) and Its Major Components

**Seiichi Matsugo** [1,2]**, Toshio Sakamoto** [3]**, Koji Wakame** [4]**, Yutaka Nakamura** [1]**, Kenichi Watanabe** [5] **and Tetsuya Konishi** [1,6,]*

[1]  Faculty of Applied Life Sciences, Niigata University of Pharmacy and Applied Life Sciences, Higashi-jima, Akiha-ku, Niigata 956-8603, Japan; matsugoh@staff.kanazawa-u.ac.jp (S.M.); nakamura@nupals.ac.jp (Y.N.)
[2]  Kanazawa University, Kakuma, Kanazawa 920-1192, Japan
[3]  School of Biological Science and Technology, College of Science and Engineering, Kanazawa University, Kakuma, Kanazawa 920-1192, Japan; tsakamot@staff.kanazawa-u.ac.jp
[4]  Faculty of Pharmaceutical Sciences, Hokkaido University of Science, Sapporo 006-8585, Japan; wakame-k@hus.ac.jp
[5]  Department of Laboratory Medicine and Clinical Epidemiology for Prevention of Noncommunicable Diseases, Niigata University Graduate School of Medical and Dental Sciences, 757, Ichiban-cho, Asahimachi-dori, Chuo-ku, Niigata 951-8510, Japan; wataken@med.niigata-u.ac.jp
[6]  Office HALD Food Function Research, Inc., Yuzawamachi Yuzawa, Niigata 949-6102, Japan
*   Correspondence: konishi@nupals.ac.jp

**Abstract:** Mibyou has been defined in traditional oriental medicine as a certain physiological condition whereby an individual is not ill but not healthy; it is also often referred to as a sub-healthy condition. In a society focused on longevity, "Mibyou-care" becomes of primary importance for healthy lifespan expenditure. Functional foods can play crucial roles in Mibyou-care; thus, the search for novel resources of functional food is an important and attractive research field. Mushrooms are the target of such studies because of their wide variety of biological functions, such as immune modulation and anti-obesity and anticancer activities, in addition to their nutritional importance. Basidiomycetes-X (BDM-X; Shirayukidake in Japanese) is a mushroom which has several attractive beneficial health functions. A metabolome analysis revealed more than 470 components of both nutritional and functional interest in BDM-X. Further isolation and purification studies on its components using radical scavenging activity and UV absorbance identified ergosterol, (10E,12Z)-octadeca-10,12-dienoic acid (CLA), 2,3-dihydro-3,5-dihydroxy-6-methyl-4H-pyran-4-one (DDMP), formyl pyrrole analogues (FPA), including 4-[2-foemyl-5-(hydroxymethyl)-1H-pyrrole-1-yl] butanamide (FPAII), adenosine and uridine as major components. Biological activities attributed to these components were related to the observed biological functions of BDM-X, which suggest that this novel mushroom is a useful resource for Mibyou-care functional foods and medicines.

**Keywords:** Basidiomycetes-X (BDM-X); Shirayukidake; *Ceraceomyces tessulatus*; Mibyou; Mibyou-care functional food; healthy lifespan expenditure

## 1. Introduction

As lifespans lengthen in developed countries, demands for a healthy lifespan are increasing, that is the life period, during which individuals are able to enjoy their health and well-being life without hospitalization or with the least amount of nursing care. Under these circumstances, "Mibyou"—a concept which originated in ancient oriental medicine—has been reevaluated, which refers to a certain physiological condition whereby an individual is not healthy but also not ill, leading to serious diagnosed endpoint diseases [1]. Mibyou-care is thus considered an important strategy for healthy life expenditure even in current preventive medicines.

According to the development of clinical examination technologies for diagnosing diseases, such as the biochemical assay of disease markers, and physical methods, including ultrasound and CT imaging, the definition of Mibyou has been updated in Western medicine as Mibyou I and II [2]. Mibyou I covers the condition in which individuals feel some unusual symptoms, such as malaise, anxiety, fatigue and pain, but clinical examinations do not show a significant abnormality. In Mibyou II, disease markers indicate certain disorders, but individuals can enjoy a normal life with the least amount of medical interventions (Figure 1).

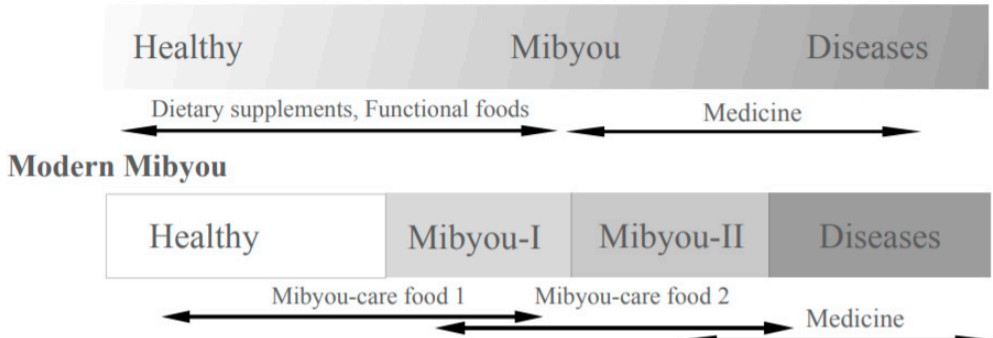

**Figure 1.** Traditional and modern Mibyou concept and Mibyou-care functional food.

The term "Mibyou-care" thus covers a wide range of practices, including daily routines such as self-medication, remedies, diet, exercise, brain storming, yoga and other activities with and without medical intervention for treating Mibyou; consequently, Mibyou-care is considered to be a basic strategy for maintaining health and well-being, both socially and individually. Functional foods and nutritional or dietary supplements obviously play important roles in Mibyou-care practices together with daily meals; thus, their roles are comprehensively discussed in the category of Mibyou-care functional food [3].

### 1.1. Mibyou-Care Functional Foods and Their Targets

Mibyou-care functional foods are broadly defined as any type of food used for Mibyou-care; that is, for disease prevention and the preservation of health and well-being. They therefore cover a wide range of food types, including nutrient-enriched fruits and vegetables and processed foods, as well as functional foods, such as food for special health use (FOSHU), nutraceuticals and medicinal food; all of these can be comprehensively grouped in the category of Mibyou-care functional foods. According to the purpose of their usage and application targets, they are currently classified into two categories: Mibyou-care functional foods 1 and 2 [3].

Mibyou-care functional food 1 is essentially used for disease prevention, health preservation and slowing the aging phenomena, and is used by healthy individuals as well as Mibyou I patients. Their major target is factors regulating metabolic homeostasis, such as immune, endocrine and neural systems, since distorted homeostatic potential is implicated as a primary condition of Mibyou in traditional oriental medicine. Foods and nutritional supplements with astringent and nourishing functions are included in this category. On the other hand, functional foods such as FOSHU and dietary supplements distributed in the current market are typical examples of Mibyou-care functional food 2. Their functions are characterized by food factors, with certain medicinal activity contributing to the treatment of Mibyou II conditions. As a result, disease markers are the target for managing respective conditions, such that levels of blood sugar, glycated proteins, blood pressure and LDL cholesterol are used to care for diabetic complications [4], and salivary AGE is suggested to be useful in the diagnosis of dementia [5]. Gsa (heterotrimeric G protein alpha) also could be a blood marker for diagnosing depression and evaluating anti-depressant drug effects [6].

Therefore, nutritional and pharmacological functions are both factors which are essential for Mibyou-care functional foods. In this context, edible mushrooms are an attractive target to be studied as a resource of Mibyou-care functional food, because they have been traditionally accepted as a food with health benefits and anti-aging functions [7].

### 1.2. Mushrooms as a Typical Resource for Mibyou-Care Functional Food

Mushrooms are not only a cuisine containing rich nutrients including vitamins and minerals [8], but they are also a medicinal resource with a variety of pharmacological functions [9,10], such as modulating immunity [11], being anticancer [12,13] and even preventing dementia [14]. Indeed, a recent meta-analysis on cohort studies indicates that mushroom consumption will reduce the risk of several diseases including cancer, and will therefore reduce the risk of mortality [15]. The social benefit of eating mushrooms has also been discussed in terms of reducing depression in the stressful conditions caused by the COVID-19 pandemic [16]. Large molecular components, such as polysaccharides including $\beta$-glucans, were primarily implicated as active principles in mushroom functions, and their anticancer function was discussed in the context of their immune modulation activity [17]. However, bioactive lower molecular weight components are also attracting significant attention, since phenylpropanoids isolated from *Inonotus obliquus (Chaga)*, for example, showed cancer cell cytotoxicity [18]. Now, a variety of components from low molecular weight compounds, such as simple phenolics, flavonoids and terpenoids, to large molecular weight components, such as polysaccharides, have been reported as the active principles of mushrooms [19–22], in addition to bioactive peptides and proteins such as ribotoxin-like protein [23,24]; their roles have been comprehensively discussed by Sanchez [25]. In addition to their nutrient composition, these bioactive ingredients characterize mushrooms as a promised resource for Mibyou-care functional food.

### 1.3. Basidiomycetes-X as a Mibyou-Care Functional Food Resource

Since mushrooms are part of the fungi family, which is a large biological kingdom [26], the search for new species of mushrooms with beneficial functions for human health is another important field of study. Basidiomycetes-X (BDM-X, Japanese name; Shirayuki-dake) is one such novel mushroom, which was originally isolated and cultivated in the mountainous district of Niigata, Japan. It was registered on the database for patented resources in Tsukuba, Ibaragi, Japan in 1999 (PCT/JP2004/006418) as a new mushroom species belonging to the Basidiomycota, which uniquely does not form mycelium. A more precise gene analysis identified BDM-X as one of the strains of *Ceraceomyces tessulatus.*

It is now artificially cultivated and provided as a cuisine, as well as a resource of functional food and medicine (Figure 2). Functional studies of BDM-X are currently progressing and have unveiled several physiological and pharmacological functions such as being anti-oxidative, anti-obesity and diabetic, and offering liver damage protection including NASH, as reviewed elsewhere [27]. The search for a functional component is also simultaneously being developed [28,29].

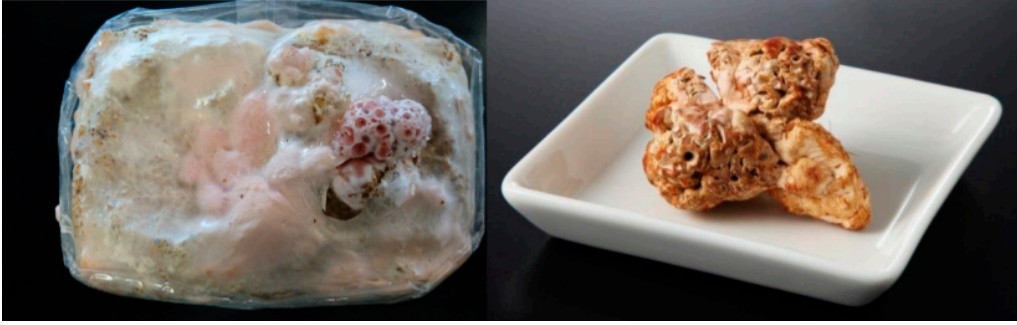

**Figure 2.** BDM-X culture and mycelium mass (provided by Mycology Techno Co. Ltd. in Niigata city, Japan).

## 1.4. Metabolome Analysis of BDM-X

To evaluate the potential of BDM-X as a functional food resource, profiling nutritionally and pharmacologically interesting components is of primary importance. Metabolomics is currently attracting attention as a method for profiling the comprehensive distribution of primary and secondary metabolites functioning in natural resources [30]. This method was applied to qualify lipophilic and hydrophilic low molecular weight organic components in BDM-X, to evaluate their nutritional and pharmacological significance. The Wide-Processed Metabolome (WPM) analysis, utilizing measurements via EC-MS and LC-MS, allowed us to detect at least 472 components (368 hydrophilic and 104 lipophilic compounds) in the BDM-X extract, which matched the migration time (MT) and mass-to-charge ratio ($m/z$) of the annotation list [31]. These include common nutrients such as amino acids, both saturated and unsaturated fatty acids, nucleotides, sugars, steroids such as ergosterol, testosterone and other secondary metabolites, including several unidentified components in addition to a variety of types of di- and tri-peptides. This wide variety of component distribution indicates the nutritional and pharmacological significance of this unique mushroom, as well as other edible mushrooms [19,23]. Notably, the metabolome analysis revealed the presence of several polyphenols, which were mainly flavonoids. The polyphenols qualitatively identified in BDM-X are as follows: 7,8-Dihydroxycoumarin, 7-Hydroxycoumarin, apigenin-7-*O*-glucoside, apigenin-8-*C*-glucoside, eriodictyol-7-*O*-neohesperidoside, eriocitrin, gallocatechin, chrysin, quercetin, delphinidin, baicalin and luteolin; their chemical structures are given in Figure 3. Polyphenols are a well-known food factor carrying anti-oxidant and anti-inflammatory functions, which play crucial roles in preventing oxidative stress-related disorders [32], and flavonoids such as apigenin and quercetin listed above are typical of them. It is generally known that the major polyphenols functioning as anti-oxidants in mushrooms are simple phenolic compounds such as phenolic acids, and flavonoids are minor since it is implicated that humans and fungi are not able to biosynthesize flavonoids [33,34]. The metabolome data suggest that the major polyphenolics in BDM-X are flavonoids, and not simple phenolic compounds. Moreover, antioxidant compounds such as ergothioneine and vitamin E, which are another group of antioxidant components found in edible mushrooms [35], were also not found in BDM-X by metabolomics and subsequent selective isolation studies [28,29]. Although the flavonoids listed above were not detected as a major ingredient in BDM-X through the isolation study, this wide variety of flavonoid distribution is unique and will significantly contribute not only to the antioxidant potential of BDM-X, but also to other physiological functions, which are both already cleared or yet to be uncovered. Further quantitative studies are required.

## 1.5. Selective Isolation and Quantification of Major Ingredients

Although the metabolome analysis identified more than 470 components of BDM-X, including a series of nutrients and the candidates of food factors, their quantitative information is limited. A further analysis of the specific components which may play significant roles in the beneficial health functions of BDM-X was carried out by solvent extraction following HPLC and TLC. The major components with UV absorption and/or DPPH radical scavenging activity were targeted for isolation, and the structures of purified compounds were assigned using mass spectrometry (MS) and nuclear magnetic resonance (NMR) [28,29]. From these studies, three types of formyl pyrrole alkaloids (FPA), ergosterol, (10*E*,12*Z*)-octadeca-10,12-dienoic acid ((10*E*,12*Z*)-CLA), 2,3-dihydro-3,5-dihydroxy-6-methyl-4*H*-pyran-4-one (DDMP), and two nucleosides, adenosine and uridine, were determined as the major ingredients existing at relatively high amounts and showing high or moderate DPPH radical scavenging activities in BDM-X. Their approximate contents determined in BDM-X are summarized in Table 1, and their chemical structures are given in Figure 4.

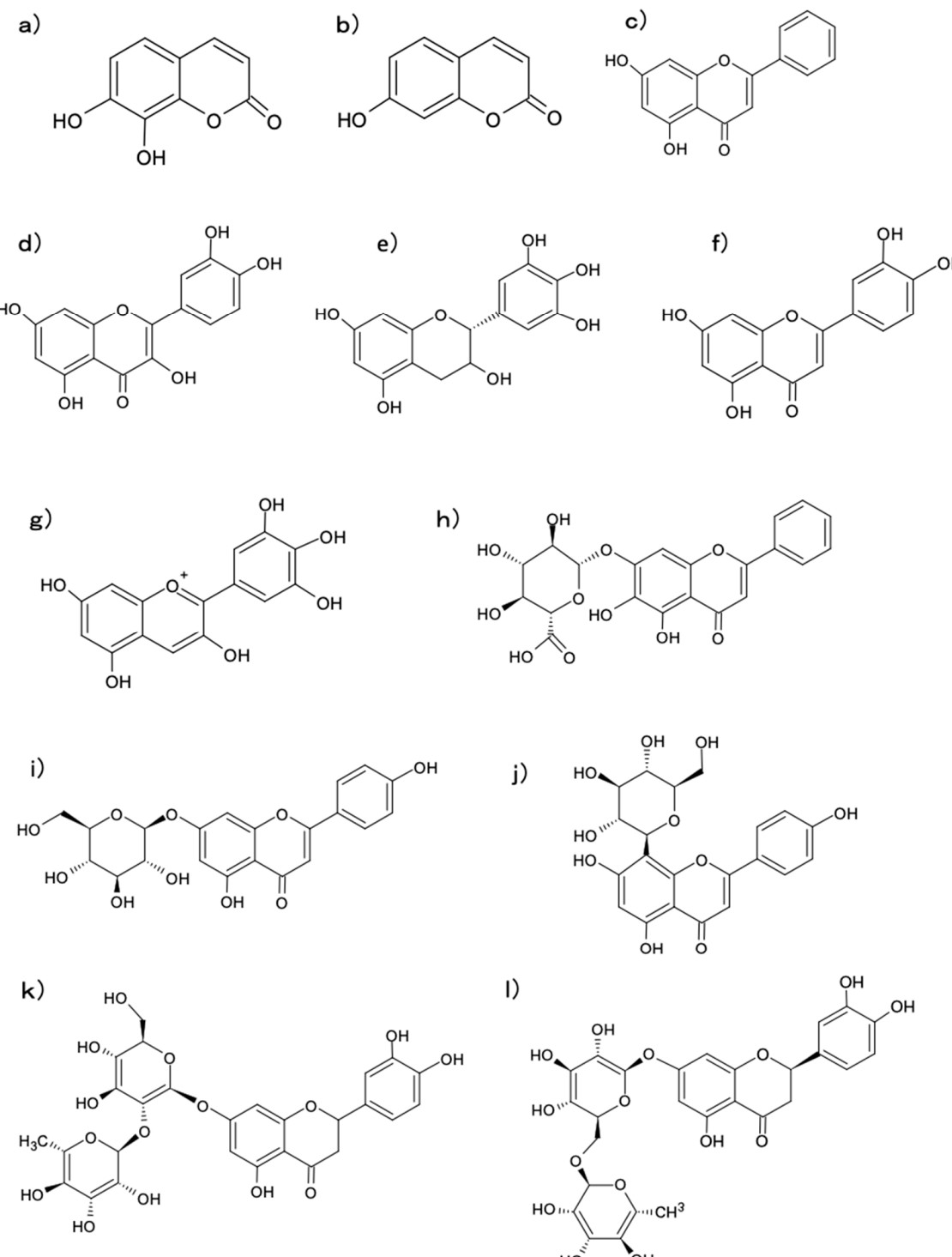

**Figure 3.** Polyphenols identified by a metabolome analysis. (**a**) 7,8-dihydroxycoumarin; (**b**) 7-hydroxycoumarin; (**c**) chrysin; (**d**) quercetin; (**e**) gallocatechin; (**f**) luteolin; (**g**) delphinidin; (**h**) baicalin; (**i**) apigenin-7-*O*-gucoside; (**j**) apigenin-8-*C*-glucoside; (**k**) eriocitrin; (**l**) eriodictyol-7-*O*-neohesperidoside.

**Table 1.** Approximate contents of major compounds identified in BDM-X.

| Compound Identified | Structure Symbol in Figure 4 | Contents (mg per 100 g BDM-X Dry Powder) | References |
|---|---|---|---|
| FPA-I | M | 82.5 | [28] |
| FPA-II | N | 48.4 | [28] |
| FPA-III | O | 1.2 | [28] |
| adenosine | P | 42.4 | [29] |
| uridine | Q | 76.9 | [29] |
| DDMP | R | 350.0 | [29] |
| ergosterol | S | 16.7 | [29] |
| (10*E*,12*Z*)-CLA | T | 19.8 | [29] |

FPA-I; 4-[2-formyl-5-(hydroxymethyl)-1*H*-pyrrole-1-yl] butanoic acid, FPA-II; 4-[2-formyl-5-(hydroxymethyl)-1*H*-pyrrole-1-yl] butanamide, FPA-III; 5-(hydroxymethyl)-1*H*-pyrrole-2-carboxaldehyde.

**Figure 4.** Structures of the major compounds identified in BDM-X (compound names are given in Table 1).

### 1.6. Role of the Major Components in Mibyou-Care Function of BDM-X

Among these identified components, formyl pyrrole analogue (FPA) carrying butyramide side chain (FPA-II) was the only new compound found in this mushroom, and is therefore a specific ingredient of BDM-X. Other compounds isolated and identified as major components of BDM-X are not new and commonly exist in many other food resources, including mushrooms. However, their nutritional and pharmacological functions suggest their pivotal role in the Mibyou-care function of BDM-X.

#### 1.6.1. FPA

FPAs are commonly distributed in a wide variety of biological resources, and their structures are also varied [36]. FPA-I and -III are found in other sources such as *Morus alba* fruits (Bilberry fruit) [37], *Inonotus obliquus (Chaga)* [38], *Lycium chinense* fruits [39,40] and *Leccinum Extremiorientale* [41]; they show a range of bioactivities including being anti-oxidant and anti-inflammation, cancer chemoprevention, macrophage activation, hepatoprotective action, and anti-obesity and anti-diabetes functions.

Since the FPA-II-carrying *N*-butyramide structure was first found in BDM-X, the FPA-II was chemically synthesized in order to assign the precise structure and to obtain enough test samples for further functional studies, as shown in Figure 5. The structure of the isolated sample assigned by spectrometry, including NMR ($^1$H, $^{13}$C 2D-NMR) and MS, was further confirmed by comparing all spectral patterns with this synthesized FPA-II. The biochemical and pharmacological functions of FPA-II therefore await clarification.

**Figure 5.** Synthetic route of newly found formyl pyrrole analogue in BDDM-X. Abbreviations: (PPTS; pyridinium *p*-toluenesulfonate, DCC; *N*,*N*′-dicyclohexylcarbodiimide, HOSu; N-hydroxysuccinimide, THF; tetrahydrofuran, EtOAc; ethyl acetate).

Since the formyl pyrrole structure (FPA-I) is chemically synthesized by the reaction of glucose and γ-butyric acid in strong acidic conditions in low yield [42], there are some discussions as to the origin of FPAs, and whether they are biosynthesized or artificially generated during the process of manufacturing dried powder. Since it is more difficult to consider the formation of FPA-bearing butyramide side chain (FPA-II) compared to FPA-I, the biosynthetic pathway of FPA-II is another target to be challenged for clarification.

### 1.6.2. DDMP

DDMP was identified as a major component with strong DPPH radical scavenging activity in the aqueous extract of BDM-X [29]. It is also isolated from several resources other than fungi, such as *lactobacterium* [43] and onion [44], and reported to have several physiological functions, such as being anti-inflammatory, anti-mutagenic and having cancer cell toxicity and affecting autonomic neurons. Tyrosinase inhibitory activity reported for DDMP is interesting because BDM-X has potential for cosmetic use [45]. DDMP, on the other hand, is implicated as a Maillard reaction product [46] which has mutagenic activity [47]. Further study is required for DDMP functioning as a food factor in BDM-X; the mechanism of biosynthetic production also awaits clarification.

### 1.6.3. Ergosterol

Ergosterol is a common component of fungal cytoplasmic membranes and plays a role in modulating membrane fluidity, similarly to the role of cholesterol in mammals; it is found in a variety of fungi or mushrooms [48]. Ergosterol is a well-known precursor of vitamin D. Ingested ergosterol is metabolized in the liver to ergocalciferol as provitamin D, and then converted to vitamins D3 with the assistance of a UV light [49]. Vitamin D is a source of hard skeletal structures such as bones and teeth, but it also contributes to the maintenance of muscle. Besides these essential roles as micronutrients, vitamin D plays a crucial role in many functions related to Mibyou-care, such as immune modulation [50], inflammation [51], hypertension and cardiovascular disease [52] and cancer [53]. A high ergosterol content in BDM-X as a vitamin D precursor suggests that BDM-X indirectly manipulates these functions.

#### 1.6.4. CLA

CLA is known as one of the essential fatty acids which is a necessary nutrient, functioning both as energy fuel and as a component of cellular membranes, as well as a biosynthetic precursor of signaling molecules [54]. Moreover, pharmacological functions of CLA are currently attracting much attention [55,56] for their anti-obesity [57], anti-carcinogenic [58], anti-hypertensive [56] and immune modulating functions [59]. There are several isomers in which 9 c,11 t-18:2 and 10 t,12 c-18:2 isomers are the most representative. Although 9 c,11 t-18:2 is the most popular isomer found in dietary substances, the major CLA found in BDM-X is the 10 t,12 c isomer. The differential function of these isomers attracts additional attention, so that both isomers have high anti-inflammatory potential different from other fatty acids, but the activity is much higher in the 10 c,12 t isomer [60].

#### 1.6.5. Adenosine and Uridine

Adenosine is a well-known nutrient molecule necessary for nucleic acids, DNA and RNA, as well as energy fuel molecule, ATP. Other phosphorylated derivatives, ADP and AMP, are metabolic intermediates acting as signaling molecules to regulate AMP kinase, which plays a pivotal role in energy homeostasis [61]. Moreover, free adenosine itself behaves as a signal molecule to modulate a variety of physiological functions related to pain, cancer and neurodegenerative, inflammatory and autoimmune diseases through interactions with adenosine receptors A1, A2A, A2B and A3 [62]. Therefore, adenosine externally taken might behave as a food factor contributing to physiological homeostasis, either directly or indirectly.

Uridine is a pyrimidine nucleoside and is a component of RNA. It is also the precursor for brain phosphatide biosynthesis, together with choline and DHA; the external administration of uridine together with DHA is therefore thought to increase synaptic proteins in the brain to protect brain aging [63].

#### 1.6.6. β-Glucan

The presence of abundant polysaccharides besides low molecular functional ingredients is one of the characteristics of fungal resources, and the role of polysaccharides has been extensively discussed in the cancer preventive function of mushrooms [64]. BDM-X also contains polysaccharides in very high amounts—approximately 33 *w/w*%—and β-glucans (13% *w/w*) are one of the characteristic sugar components of BDM-X [27]. It is known that β-glucans play a critical role in the inert immune modulating activity of mushrooms as pathogen-associated molecular pattern molecules (PAMPs) to stimulate toll-like receptor 2 (TLR2) [65]. The possible application of BDM-X as a high β-glucan resource has previously been discussed in relation to the immune modulator used during the COVID-19 pandemic [66]. Preventing obesity is another important function of β-glucans as a source of dietary fiber [67].

### 1.7. Medicinal and Pharmacological Functions of BDM-X Contributing to Mibyou-Care

So far, several animal and human studies on the medicinal functions of BDM-X have been carried out and are summarized in Table 2; these include antioxidant protection [68], anti-obesity effects [69,70], hepatoprotective functions [69,71,72] and immune modulation, including the ameliorative effect on atopic dermatitis [73,74]. It is worth discussing the possible contribution of identified BDM-X components to those already known, as well as other unpublished functions.

**Table 2.** Reported biological and pharmacological functions of BDM-X.

| | Medicinal Functions | Exp. System | Test Sample Form | Experimental and Results | Refs. |
|---|---|---|---|---|---|
| 1 | Antioxidant activity | in vitro, in situ | Aqueous extract | BDM-X prevented AAPH induced peroxidation in rat liver homogenate. Pre-administration of BDM-X to rat prevented nitrotyrosine formation followed LPS induced liver injury. | [68] |
| 2 | Anti-obesity, anti-diabetic and liver protective function | Male albino rat and OLETOF rat | BDM-X powder and extracts | BDM-X supplementation suppressed weight gain, visceral fat deposit and fatty liver injury caused by 15 weeks feeding on an HFHS diet, and ameliorated insulin sensitivity and adiponectin expression. | [69,70] |
| 3 | Amelioration of atopic dermatitis in humans | Human | BDM-X powder | Oral intake of BDM-X powder for two weeks ameliorated atopic dermatitis symptoms in volunteers. | [74] |
| 4 | Alleviation of atopic dermatitis in mice | Mouse | BDM-X powder | BDM-X administration to atopic dermatitis (AD) induced by house dust mite extract application in NC/Nga mouse attenuated ADlike clinical symptoms through modulating Th1/Th2 responses. | [73] |
| 5 | Prevention of nonalcoholic steatohepatitis (NASH) | Mouse | BDM-X powder | In NASH-HCC mice (C57BL/6J female pups) model produced by STZ-high fat diet treatment, BDM-X prevented pathogenesis of NASH by preventing inflammation and lipogenesis. | [71,72] |
| 6 | Hepatoprotective function | Human | BDM-X powder | The effect and safety of BDM-X on fatty liver were evaluated by a stratified randomized double-blind parallel group comparison. | [75] |

Abbreviations: AAPH; 2,2′-azobis (2-amindino-propane) dihydrochloride, PS; lipopolysaccharide, HFHS diet; high fat, high sucrose diet, STZ; streptozotocin, HCC; hepatocellular carcinoma.

1.7.1. Protection against Oxidative Stress and Inflammation

Antioxidant and anti-inflammatory activities are a basic requirement of Mibyou-care functional foods, because they commonly occur in life processes to distort metabolic homeostasis, and are implicated as a causative factor of not only aging deterioration, but also pathogenesis and the progression of many diseases [33,76].

BDM-X has high potential for antioxidant or free radical scavenging activities, especially against the hydroxyl radical, which is the ultimate reactive species damaging cellular components including DNA by its hydrogen abstraction mechanism [77]. There are many antioxidant molecules which have been reported as quenching the hydroxyl radical effectively in vitro, but a few are also active in vivo. However, BDM-X has effective hydroxyl radical scavenging potential in vivo, too; it was previously shown that orally given BDM-X effectively prevented lipopolysaccharide induced liver damage in rodents, and nitrotyrosine formation, which is the marker of the hydroxyl radical induced damage, was also inhibited [68]. This early observation on the hydroxyl radical scavenging potential of BDM-X is rationalized by the action of major BDM-X ingredients listed in Table 2,

especially FPAs, CLA, ergosterol and nucleotides, because they have the hydrogen atom as a target of hydroxyl radical attack. Indeed, the hydroxyl radical scavenging and cancer chemo preventive activity of Goji berry originated FPA-I has been reported [40].

Besides the hydroxyl radical scavenging activity, the high antioxidant potential of BDM-X has been proved by several other assay methods in vitro, such as DPPH radical scavenging activity, $Fe^{3+}$-reducing ability, $Cu^{2+}$-reducing ability and $Fe^{2+}$-chelating activity [78]. The total phenolic content in the aqueous extract determined elsewhere was as high as 8.1 mg gallic acid equivalent/g dry powder, and this value is comparable with the values reported for edible anti-oxidative mushrooms from Poland (3–12.8 mg gallic acid/g) [79]. This indicates that a series of polyphenols, mainly flavonoids, determined by the metabolome analysis (Figure 3) will also provide a significant contribution to the antioxidant potential of BDM-X. Since the oxidative damages are mediated by the reactive species with diverse reactivity and cellular localization in vivo, the synergistic functions of antioxidant components with different reactivity are also implicated in the high antioxidant potential of BDM-X.

### 1.7.2. Anti-Obesity and Anti-Metabolic Syndromes Function of BDM-X

Obesity is one of the major targets of Mibyou-care, because it is the major pathogenic condition of diabetes and related diseases including cardiac failure, stroke, dementia and cancer [80]. BDM-X has been shown to have a marked anti-obesity function in rodents, whereby male albino rats and genetically obese rats (OLETF) were fed a high-fat/high-sucrose (HFHS) diet with and without DBM-X supplementation for 90 days [69,70]. BDM-X supplementation markedly inhibited body weight gain, reduced visceral fat deposits and improved insulin tolerance acquired by the HFHS diet. Supporting this observation, anti-obesity and anti-diabetic functions have been studied for FPAs [36] and CLA [57], which were detected as the major components of BDM-X in addition to $\beta$-glucan as dietary fiber [67]. Polyphenols, typically catechins, have also been reported for their anti-obesity function through the manipulation of lipid metabolism [81]. Moreover, the high antioxidant and inflammation activities of BDM-X sustained by these antioxidant components including polyphenols [82] will obviously contribute to the regulation of insulin sensitivity, as was observed in HFHS feeding experiments [69,70].

### 1.7.3. Hepatoprotective Function of BDM-X

A strong hepatoprotective effect is one of the attractive pharmacological functions of BDM-X. Long-term feeding of an HFHS diet produced a fatty liver and increased the level of transaminases as the liver injury marker in rats, but BDM-X supplementation in the diet suppressed these changes [69], indicating their hepatoprotective function. The liver-protective function of BDM-X was further studied in the rodent model of non-alcoholic steatohepatitis (NASH), where NASH was induced by streptozotocin (STZ) in combination with feeding an HFHS diet [71,72]. NASH is the chronic liver disease regarded as Mibyou II, which can lead to liver cancer, and is closely associated with obesity and diabetes [83]. Gavage administration of BDM-X to mice in the nonalcoholic fatty liver stage (NAFLD) effectively prevented disease progression into NASH and inhibited fibrosis (Figure 6). The dietary supplemented BDM-X attenuated an enhanced expression of sterol regulatory element binding protein isoform (SREBP-I) and peroxisome proliferator-activated receptors (PPAR-alfa) during the development of NASH, indicating that BDM-X primarily manipulates lipogenesis in the liver [72]. The finding that BDM-X enhanced adiponectin expression in obese rats also suggests that BDM-X manipulates lipid metabolism, which leads to the prevention of obesity and fatty liver formation [70]. The ameliorative effect of BDM-X on fatty liver was also currently evaluated in humans by a randomized, double-blind, parallel-group comparison study, and a significant improvement of aminotransferase enzyme level was observed [75].

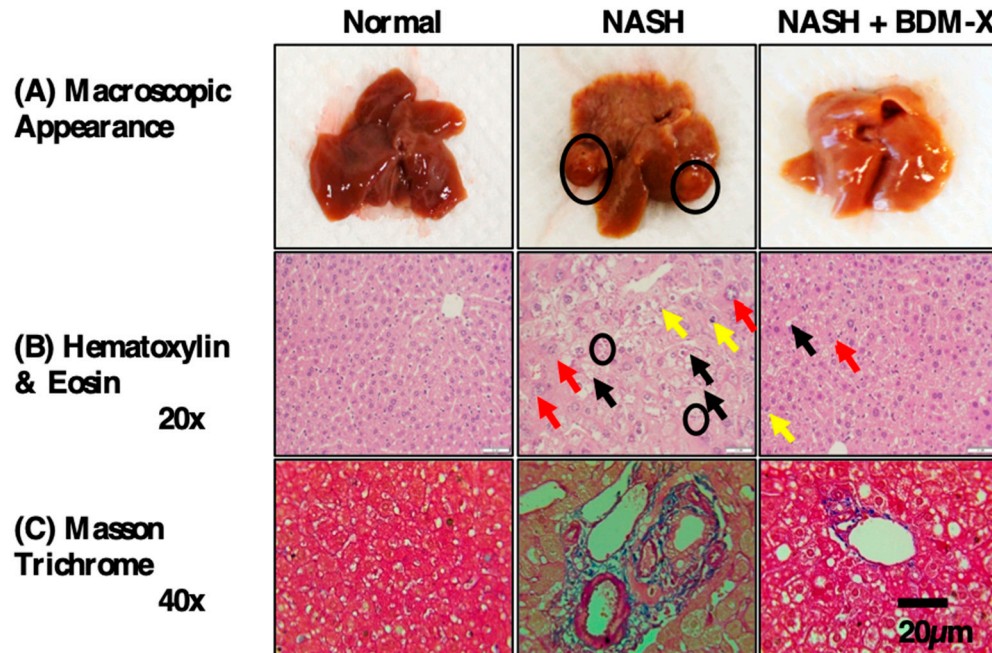

**Figure 6.** BDM-X attenuates clinicopathology in NASH-HCC mice. (**A**) Representative macroscopic appearance of livers (circles: liver tumors). (**B**) Hematoxylin/eosin staining (black arrow: macrovesicular steatosis, yellow arrow: microvesicular steatosis, red arrow: hypertrophy, circles: inflammatory cells). (**C**) Fibrosis deposition by Masson's trichrome staining (blue area). Normal, age-matched mice subjected to a normal diet; NASH, streptozotocin injected mice subjected to being fed a high-fat diet up to 16 weeks of age; NASH+BDM-X; streptozotocin injected mice subjected to the high-fat diet, treated with BDM-X (500 mg/Kg/day) from the age of 12 weeks to 16 weeks. Scale bar = 20 μm.

Liver disorders include NASH associated with oxidative stress and inflammation. The antioxidant components of BDM-X, including CLA, especially 10 c, 12 t CLA, ergosterol and adenosine, also convey anti-inflammatory activities and are therefore expected to contribute to the hepatoprotective function of BDM-X. FPA-I has the hepatoprotective function which reportedly carries anti-inflammatory and anti-oxidative stress activities [38,39]. It is also important to note that polyphenols, including flavonoids typically, display anti-inflammatory activities as well as anti-oxidant activities [84]. Indeed, the hepatoprotective function of flavonoids such as quercetin and naringin, for example, have been reported [85,86]. Therefore, the comprehensive and synergistic actions by these BDM-X components will be reflected in the effective prevention and amelioration of inflammation disorders, and typically liver damage diseases including NASH.

### 1.7.4. Immune Modulating Function

Regarding homeostasis regulation, the immune system is of primary importance [87] and is therefore the target of Mibyou-care, especially Mibyou I. Mushrooms have attracted attention as immune modulators because of their rich nutrients, encompassing both major and micro-nutrients, and also due to the presence of food factors, which modulate immune cell activity [88]. The immune activating function of polysaccharides, especially β-glucan, has been discussed mainly in relation to cancer therapy [89], and certain polysaccharide fractions such as Krestin are clinically used as medication to treat cancer [90].

BDM-X is also predicted to have immune modulation activity because of the high contents of β-glucan (13% *w*/*w*). Other components identified in BDM-X, especially FPAs and CLA, are reported to modulate immune cell activation [39,59]. Adenosine acts as an endogenous modulator of inert immunity, which plays a crucial role in Mibyou I care [91]. Ergosterol is another component, probably indirectly modulating the immune system through the formation of Vitamin D as an immunity modulator [50]. The effects associated

with these major components may explain the observed functions of BDM-X, such as inhibition and amelioration of atopic dermatitis reported both in rodent [73] and human studies [74], where antioxidant and anti-inflammatory functions also have cooperative roles. Indeed, histochemical observation showed that BDM-X manipulated the accumulation of inflammatory must cells in the damaged skin of the atopic dermatitis model mice (Figure 7).

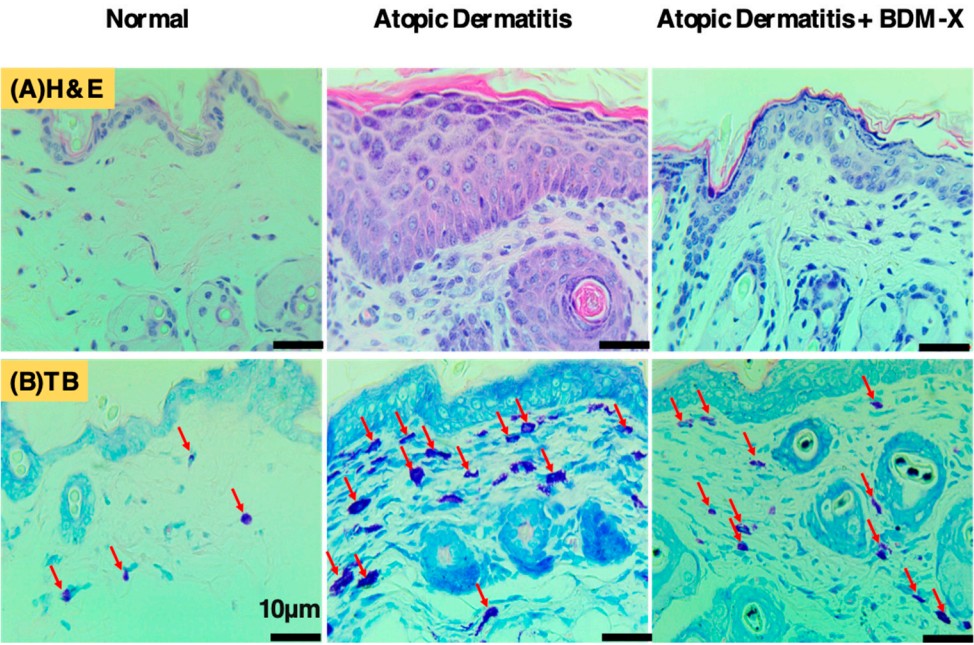

**Figure 7.** BDM-X treatment improves histopathological changes in atopic dermatitis in mice. (**A**) Hematoxylin/eosin (H&E) staining of the cross-sectional tissue slices of skin showing hyperkeratosis, parakeratosis, acanthosis and spongiosis. (**B**) Skin levels of mast cells (red arrow) by toluidine blue (TB) staining. Scale bar = 10 μm.

### 1.7.5. Cancer Preventive Function

The cancer immune activation is another aspect of the BDM-X function. There are a few preliminary unpublished observations, such that BDM-X administration stimulated lymphocyte formation in the spleen of rodents and increased cancer cell specific lymphocytes in the blood of a stage IV cancer patient. β-glucan is implicated as an active principle, but the contribution of lower molecular weight components determined in BDM-X is also plausible, since the chemo preventive function against cancer cells is reported for FPAs [36,40], CLA [58,92] and DDMP [44,93]. More precise studies are required before establishing the possible application of BDM-X in cancer prevention as a functional food to stimulate cancer immunity in Mibyou-care routines [88], as well as in the treatment of cancer in complimentary medicine as an adjuvant in chemo- and radio-therapies.

### 1.7.6. Other Prospective Functions

From the reported bioactivities of the respective components discussed above, BDM-X is an attractive research target for the study of functions such as blood pressure control, cardiovascular disease and the prevention of neuronal diseases, including dementia. For example, a BDM-X component, especially CLA, having high antioxidant and anti-inflammatory potential is reported to prevent neuroinflammatory conditions leading to brain damage [94].

There are currently many disorders including those described above, which are extensively discussed in relation to microbiome [95–97], and the β-glucans are the typical ingredient which can modulate intestinal bacterial flora [98]. However, the effect of other BDM-X components on intestinal bacteria also attracts attention besides their conventional pharmacological effects. We preliminarily observed that the long-term ingestion of BDM-X

in rodents affected intestinal microbiome, so as to decrease certain bacterial families such as *Allobaculum*, which is associated with obesity, and to increase *Bacteroides* (not published). Therefore, further studies are warranted to clarify the effects of BDM-X and ingredients on gut bacteria.

*1.8. Characteristic Feature of BDM-X as Mibyou-Care Functional Food*

Among the major components of BDM-X, FPAs and DDMP are xenobiotics, but others are physiological substances. Pharmacologically interesting food factors are commonly xenobiotics such as polyphenols and dietary fibers, but it turns out that some cellular or physiological components show certain pharmacological functions, and are therefore called metabolic intermediate-type food factors, as exemplified by squalene [99]. Xenobiotics as a pharmacologically active substance obviously contribute to observed food functions, but latter-type food factors also play pivotal roles, especially in Mibyou-care, as shown by examples such as branched amino acids, especially leucine, which modulate skeletal muscle remodeling through attenuating inflammation [100] and Omega-3 fatty acids, especially EPA, which behave as an anti-inflammatory substance to prevent neuronal diseases [101]. Similarly, adenosine and uridine found in BDM-X as the major component make a significant contribution as metabolic intermediate-type food factors in the Mibyou-care functions of BDM-X.

Mibyou-care also involves medical intervention, especially in Mibyou II care. Although the mechanism of food factor action as a pharmacologically active molecule is rationalized by ligand–receptor interaction, the specificity and binding strength of food factors are generally weak compared to medicine, and strong pharmacological activity is therefore not the primary requirement of food factors [102]. Metabolic intermediate-type food factors, including adenosine and uridine as signaling molecules [103], might play a pivotal role in the homeostatic regulation of physiological reactions, which is one of the basic targets of Mibyou-care. In this context, the biological response modifier (BRM) effect that was primarily implicated for the anticancer function of mushrooms [104] might be important as an underlying mechanism of the Mibyou-care function of BDM-X.

Although the pharmacological contributions of respective components have not been precisely studied yet, comprehensive action, including synergism among the components, is considered to play a pivotal role in the beneficial health functions of BDM-X as an edible mushroom, and characterizing the potential of BDM-X as a Mibyou-care functional food.

## 2. Conclusions

The major components identified in BDM-X strongly indicate that BDM-X has attractive properties, which may play significant roles in Mibyou-care practice as a cuisine and also as functional food resources. Its medicinal application should also be further discussed elsewhere.

Further mechanistic studies are needed at a molecular level to understand how the isolated components, especially FPA-II, are involved in the observed functions of BDM-X. However, the discussion above indicates that BDM-X itself is a promising food applicable to Mibyou-care, especially Mibyou I care.

**Author Contributions:** Conceptualization, T.K.; validation, S.M., T.S., K.W. (Koji Wakame), K.W. (Kenichi Watanabe), and Y.N.; resources, T.K., S.M., T.S., K.W. (Koji Wakame), K.W. (Kenichi Watanabe). and Y.N.; data curation, T.S., K.W. (Koji Wakame), K.W. (Kenichi Watanabe). and Y.N.; writing—original draft preparation, T.K.; writing—review and editing, T.K., S.M., T.S., K.W. (Koji Wakame). and Y.N.; visualization, K.W. (Koji Wakame), K.W. (Kenichi Watanabe). and Y.N.; project administration, T.K.; funding acquisition, T.K. All authors have read and agreed to the published version of the manuscript.

**Funding:** This review paper is not funded by any profit or organization. Original studies by coauthors sited here were funded by several sources of grant including the grant from the Ministry of Education, Culture, Sports, Science and Technology of Japan for K. Watanabe (23602012), and from the Promotion and Mutual Aid Corporation for Private Schools, Japan for K. Watanabe (26460239) and also T.K., and

from the Joint research contract with Kanazawa University and Mycology Techno Co., Ltd. (Niigata, Japan) for T. S. These are described in their original papers published.

**Institutional Review Board Statement:** None of the data presented in this study required approval from any institution or board.

**Informed Consent Statement:** There are no descriptions in this study which need informed consent agreement.

**Data Availability Statement:** None.

**Conflicts of Interest:** The authors declare no conflict of interest.

## Abbreviations

| | |
|---|---|
| CT | Computed Tomography |
| HPLC | High-Performance Liquid Chromatography |
| GC | Gas Chromatography |
| GC-MS | Gas Chromatography-Mass Spectroscopy |
| EC-MS | Electrochemical Mass Spectroscopy |
| LC-MS | Liquid Chromatography Mass Spectroscopy |
| NMR | Nuclear Magnetic Resonance |
| UV | Ultraviolet |
| DPPH | 2,2-Diphenyl-1-Dipicryl Hydrazyl Radical |

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
