# Peer review of "Mushrooms as a Resource for Mibyou-Care Functional Food; The Role of Basidiomycetes-X (Shirayukidake) and Its Major Components"

_nutraceuticals, doi:10.3390/nutraceuticals2030010_

Round 1

Reviewer 1 Report

In the review entitled ‘Mushrooms as the Resource for Mibyou-care Functional Food; The Role of Basidiomycetes-X (Shirayukidake) and Its Major Ingredients as the Example’ the authors analyse the functional foods for the crucial roles in the Mibyou-care, a physiological condi-18 tion, neither healthy nor sick, but very important due to the life span elongation. In particular, the authors pay attention to mushrooms as typical resource for Mibyou-care functional food describing the Basidiomycetes-X (BDM-X, Japanese name; Shirayukidake). The review is of interest, however some inconsistencies need to be solved before possible publication on nutraceuticals journal, as reported below:

-I suggest to revise the Title by shortening it and deleting the term ‘as the Example’. Avoid the term ‘ingredient’;

-report the affiliation numbers near authors name as apex;

- I would have expected in the introduction, some reference on mushrooms as source of several bioactive compounds highly studied and reviewed. Indeed, edible basidiomycetes mushrooms are a rich source of bioactive compounds such as small molecules, polysaccharides, proteins, and polysaccharide–protein complexes. It is important to include that mushrooms are a reservoir of enzymes like ribotoxin-like proteins with therapeutic potential or bioactive peptides with several important biological properties. On this regard, in literature authors can find some recent reviews (e.g. Toxins (Basel). 2021 Apr; 13(4): 263 for proteins and Toxins (Basel). 2022 Jan 22;14(2):84 for peptides). Please revise the literature accordingly. This is also for paragraph 1.2.

-Check the typos in the text and revise in overall text (e. g. page 1, line 39 ‘hos-pitalization’ and 40 ‘an-cient’);

-please avoid the term ‘ingredient’ when referring to flavonoids (line 155) or other molecules (line 181 and so on)

-Write Figure legend in text form used for the manuscript, not like a Figure caption and improve the quality/resolution of images, especially Fig. 6.

-Write both Tables legend and notes in text form used for the manuscript. It is better to report also Table in text form to improve the quality and resolution

Please revise the references style according to the journal guidelines:

e.g. Author 1, A.B.; Author 2, C.D. Title of the article. Abbreviated Journal Name Year, Volume, page range.

Note that the year is in bold.

Moreover, why the references in the main text are reported in red?

Author Response

Thank you for your reviewing of our manuscript. 

We carefully pursued reviewer’s comments and suggestions, and these are sound quite 

Reasonable. Therefore reversions are basically made according to their suggestion.

Please find that the parts that revision was made are indicated by highlighting yellow in the revised version. 

Table and Figure titles and legends are separately prepared and set between conclusion and references section. 

The figures and Tables are drown in the separate file, because it was not easy to insert in the text at right position.  Also please find one figure as Fig. 7 was newly added. 

Specific responses to the reviewer’s comments are attached below. 

Sincerely, 

Tetsuya Konishi  

Specific response to respective comment.

-I suggest to revise the Title by shortening it and deleting the term ‘as the Example’. Avoid the term ‘ingredient’;

→ The title was changed according to this suggestion.

-report the affiliation numbers near authors name as apex;

→ corrected.

- I would have expected in the introduction, some reference on mushrooms as source of several bioactive compounds highly studied and reviewed. Indeed, edible basidiomycetes mushrooms are a rich source of bioactive compounds such as small molecules, polysaccharides, proteins, and polysaccharide–protein complexes. It is important to include that mushrooms are a reservoir of enzymes like ribotoxin-like proteins with therapeutic potential or bioactive peptides with several important biological properties. On this regard, in literature authors can find some recent reviews (e.g. Toxins (Basel). 2021 Apr; 13(4): 263 for proteins and Toxins (Basel). 2022 Jan 22;14(2):84 for peptides). Please revise the literature accordingly. This is also for paragraph 1.2.

→ several recent review on the functional components of mushrooms are cited to the reference list, and accordingly certain revision and addition are also made in paragraph1.2.

-Check the typos in the text and revise in overall text (e. g. page 1, line 39 ‘hos-pitalization’ and 40 ‘an-cient’);

→ corrected these and other part as much as possible.

-please avoid the term ‘ingredient’ when referring to flavonoids (line 155) or other molecules (line 181 and so on)

→ substituted the term ingredient to other terms such as component or compound as possible. However, the term ingredient is used frequently for the component substances in food and other natural resources, the term ingredient is yet used in several other parts.

-Write Figure legend in text form used for the manuscript, not like a Figure caption and improve the quality/resolution of images, especially Fig. 6.

  Figure legends are separately prepared in text format and the Fig. 6 was also revised so as to clear the figure and captions in the figure.

-Write both Tables legend and notes in text form used for the manuscript. It is better to report also Table in text form to improve the quality and resolution

→ Fig and Table legends are separately prepared and attached in front of reference section.

Please revise the references style according to the journal guidelines:

e.g. Author 1, A.B.; Author 2, C.D. Title of the article. Abbreviated Journal Name Year, Volume, page range.

Note that the year is in bold.

Moreover, why the references in the main text are reported in red?

→ reference format was revised according to the journal guideline. Sorry, the red typing of reference number in the previous manuscript does not mean any special meaning but just for avoiding confusion in the process of manuscript preparation. 

Reviewer 2 Report

In this review, unique Basidiomycetes-X (Shirayukidake) mushroom was introduced and its main components and bioactivities were summarized, for potential application in sub-health care. The review topic is very interesting, and the notion of Mibyou-care and the relevant functional food ingredients are valid and attractive to readers in nutraceutical community. However, major improvement in structure/organization of contents and the quality of tables is required for further consideration. Some other points below are also need to be corrected or addressed:  

Line 29: pyrrole alkaloids

Line 184: fruit of Morus (mulberry fruit), should add the Latin name; also the Latin name should be given to Chaga as that for other plants mentioned

Line 191: 13C, 2D-NMR

Table 1: The content and yield listed for major components are not in agreement. It is understandable that isolation yield will be lower than content (determined by analytical measurement), but the data presented in Table 1 are quite the opposite.

Line 203-267: Would section 1.8 to 1.12 be part of section 1.7? If that is case, it should be listed as 1.7.1 to 1.7.5, not 1.8 to 1.12. Note that line 268 is with the heading of 1.12 too. 

Line 256: β-Glucan 

Line 277-395: Again, the section heading numbers are confusing. Sections 1.13 to 1.18 should be 1.12.1 to 1.12.6.

Author Response

Thank you for your reviewing of our manuscript. 

We carefully pursued reviewer’s comments and suggestions, and these are sound quite 

Reasonable. Therefore reversions are basically made according to their suggestion.

Please find that the parts that revision was made are indicated by highlighting yellow in the revised version. 

Table and Figure titles and legends are separately prepared and set between conclusion and references section. 

The figures and Tables are drown in the separate file, because it was not easy to insert in the text at right position.  Also please find one figure as Fig. 7 was newly added. 

Specific responses to the reviewer’s comments are attached below. 

Sincerely, 

Tetsuya Konishi  

Responses to respective comments.

Line 29: pyrrole alkaloids

→ revised to formyl pyrrole analogues

Line 184: fruit of Morus (mulberry fruit), should add the Latin name; also, the Latin name should be given to Chaga as that for other plants mentioned

→ corrected and added Latin names according to your suggestion.

Line 191: 13C, 2D-NMR

→ corrected.

Table 1: The content and yield listed for major components are not in agreement. It is understandable that isolation yield will be lower than content (determined by analytical measurement), but the data presented in Table 1 are quite the opposite.

→ Primary, the yield was given to show the reliability of isolation process, but this gives some confusion and thus the yield was delated from revised Table I. Since the extracts prepared by different solvent was used for starting extracts for isolation, yield may not exactly reflect in the amounts finally calculated.

Line 203-267: Would section 1.8 to 1.12 be part of section 1.7? If that is case, it should be listed as 1.7.1 to 1.7.5, not 1.8 to 1.12. Note that line 268 is with the heading of 1.12 too. 

→ The sections were rearranged and section numbers were corrected so as to make discussions clearer.

Line 256: β-Glucan 

→ corrected.

Line 277-395: Again, the section heading numbers are confusing. Sections 1.13 to 1.18 should be 1.12.1 to 1.12.6.

→ as mentioned above, sections were rearranged. 

Round 2

Reviewer 1 Report

no comment

Author Response

We carefully reviewed the English writing and corrected several typos, grammatical errors and unclear sentences as much as possible. Revised parts are indicated by yellow background.

Reviewer 2 Report

The revised version improved the manuscript quality. A few minor points to be addressed:

1. Line 185: Table 1. The title "Approximate Contents of Major Compounds identified in BDM-X". Abbreviations such as FPA-1... should be put at the bottom of table. Table quality (as image) is low. Also, "Contents determined per 100 mg BDM-X dry powder", would that be microgram? weight unit is missing.

2. Line 290. Table 2. Quality is low (image), also please put abbreviations at the bottom of table.

From the Editorial Office: Please provide editable tables.

Author Response

We carefully reviewed the English writing and corrected several typos, grammatical errors and unclear sentences as much as possible. Revised parts are indicated by yellow background.

  1. Line 185: Table 1. The title "Approximate Contents of Major Compounds identified in BDM-X". Abbreviations such as FPA-1... should be put at the bottom of table. Table quality (as image) is low. Also, "Contents determined per 100 mg BDM-X dry powder", would that be microgram? weight unit is missing.

  Thank you for your indication of our careless mistake. We corrected both title and tables. And the revised tables are given in the editable form.

2. Line 290. Table 2. Quality is low (image), also please put abbreviations at the bottom of table.

→  Tables are given in the editable form, and abbreviations are given at the bottom of each table.